# Chronic Systemic SARS-CoV-2 Infection Without Respiratory Involvement in an Immunocompromised Patient

**DOI:** 10.3390/v17020147

**Published:** 2025-01-23

**Authors:** Francisco Tejerina, Daniel Peñas-Utrilla, Marta Herranz, Pilar Catalán, Mercedes Marín, Lara Mesones, José Manuel García-Domínguez, Beatriz Merino, Leire Pérez, Chiara Fanciulli, Patricia Muñoz, Carmen Rodríguez-Gonzalez, Cristina Diez, Teresa Aldámiz, Andrea Molero-Salinas, Laura Pérez-Lago, Darío García de Viedma

**Affiliations:** 1Clinical Microbiology and Infectious Diseases Department, Hospital General Universitario Gregorio Marañón, 28007 Madrid, Spain; daniel.penasutrilla98@gmail.com (D.P.-U.); m_herranz01@hotmail.com (M.H.); pilar.catalan@salud.madrid.org (P.C.); mercedes.marinar@salud.madrid.org (M.M.); laramesonesguerra@gmail.com (L.M.); legor78@hotmail.com (L.P.); fanciulli.chiara@gmail.com (C.F.); pmunoz@hggm.es (P.M.); crispu82@gmail.com (C.D.); teresaldamiz@yahoo.es (T.A.); andrea.molero@iisgm.com (A.M.-S.); dgviedma2@gmail.com (D.G.d.V.); 2Instituto de Investigación Sanitaria Gregorio Marañón, 28009 Madrid, Spain; lperezg00@gmail.com; 3CIBERINFEC, CIBER de Enfermedades Infecciosas, Instituto de Salud Carlos III, 28029 Madrid, Spain; 4Department of Neurology, Hospital General Universitario Gregorio Marañón, 28007 Madrid, Spain; jose_garciadom@hotmail.com; 5Gastroenterology Department, Hospital General Universitario Gregorio Marañón, 28007 Madrid, Spain; beatriz.merino@salud.madrid.org; 6Departamento de Medicina, Facultad de Medicina, Universidad Complutense de Madrid, 28040 Madrid, Spain; 7CIBERES, CIBER de Enfermedades Respiratorias, Instituto de Salud Carlos III, 28029 Madrid, Spain; 8Pharmacy Department, Hospital General Universitario Gregorio Marañón, 28007 Madrid, Spain; crgonzalez@salud.madrid.org

**Keywords:** persistent COVID-19, immunocompromised, systemic infection

## Abstract

In a patient on immunosuppressant treatment, SARS-CoV-2 RNA was documented in different extra-respiratory samples over several months in the absence of positive determinations in upper respiratory samples. Whole-genome sequencing of these samples showed the acquisition of different single-nucleotide polymorphisms over time, suggesting viral evolution and thus viral viability.

## 1. Introduction

SARS-CoV-2 virus is a betacoronavirus responsible for COVID-19. Few months after the beginning of the pandemic, different reports started to describe that some immunocompromised patients presented persistent infections with recurrent pulmonary infiltrates that could be the cause of the death several months after the initial infection [1]. These patients maintained positive determinations of SARS-CoV-2 RT-PCR in respiratory specimens with documentation of virus viability, and in some cases, viral persistence could last for months [2]. In these patients, SARS-CoV-2 can also be detected in extra-respiratory samples and tissues. 

Attempts to define persistent SARS-CoV-2 infection in immunocompromised patients have been made, based on microbiological, clinical, and radiological criteria. Microbiological criteria are focused on screening for viral presence in respiratory samples and not extra-respiratory samples. Different reports have evaluated the administration of combined antiviral treatment to these patients, achieving viral clearance in a relevant proportion of these patients [3].

A different syndrome denominated Long COVID develops in non-immunocompromised patients after COVID-19. This syndrome resembles a chronic fatigue syndrome, and one of the proposed casual hypotheses associated with the development of this clinical syndrome is SARS-CoV-2 viral persistence, although, actually, this hypothesis has not been validated.

SARS-CoV-2 should be considered a systemic virus with a particular respiratory tropism that could also establish systemic chronic infection in immunocompromised patients in the absence of documented viral infection in nasopharyngeal samples [4]. 

We report the presence of SARS-CoV-2 RNA in different extra-respiratory specimens in an immunocompromised patient without detection of virus in upper respiratory samples and without pneumonia or respiratory symptoms. Whole-genome sequencing of different samples taken over several months showed the progressive acquisition of several single-nucleotide polymorphism (SNPs), suggesting viral evolution in extra-respiratory sites.

## 2. Case Report

### 2.1. Patient History

A 67-year-old man with a previous diagnosis of chronic recurrent idiopathic optic neuritis receiving immunosuppressant treatment was first evaluated in our outpatient infectious diseases clinic, Hospital General Universitario Gregorio Marañón, on 11 February 2022. 

The patient had been diagnosed with COVID-19 in March 2020 by means of RT-PCR in a nasopharyngeal sample. Clinical curse of the patient was mild, presenting fever and upper respiratory symptoms that resolved in a few weeks without need for hospitalization. After this episode, the patient started with fatigue, muscle pain, dyspnea, dyspepsia, post-prandial fullness, and dizziness. These symptoms fluctuated in intensity but maintained over time and fulfilled the criteria of Long COVID.

In February 2021, the patient was diagnosed with bilateral optic neuritis at another clinical center. Cervical and cranial nuclear magnetic resonance, immunological assays, and cerebrospinal fluid (CSF) analysis were normal, there were no microbiological findings, and anti-aquaporin and anti-myelin oligodendrocyte glycoprotein antibodies were negative. The patient received high-dose methylprednisolone, plasmapheresis, and steroid treatment with prednisone at 30 mg/day with partial recovery. In September 2021, the patient started again with progressive visual loss, receiving a new steroid bolus and treatment with azathioprine 100 mgs/day.

There were no significant records in his personal history apart from the presence of two subcentimeter lung nodules, followed-up at another center in a context of lung cancer screening. The patient did not report SARS-CoV-2 re-infections since 2020. Thoraco-abdominal computed tomography (CT) performed at our center showed two lung nodules of approximately 7 mm similar to previous CT, with no other findings. A blood analyses we performed showed a severe cellular and humoral immunosuppression (Table 1).

SARS-CoV-2 whole-blood RT-PCR was positive, with a cycle threshold (Ct) of 38, and negative on NP specimens. Five days later, the same results were obtained (Table 2). The patient denied symptoms compatible with COVID-19 in the previous months and had received three doses of SARS-CoV-2 vaccine BNT162b2 in 2021.

In April 2022, due to the gastrointestinal symptoms reported by the patient, a gastroscopy was performed. Tissue biopsies of gastric body and antrum showed no relevant histological findings, without atrophy, metaplasia, or dysplasia. SARS-CoV-2 RT-PCR of the gastric biopsies was positive.

In June 2022, whole-blood and -stool samples were positive for SARS-CoV-2 RNA, and an RT-PCR conducted on cerebrospinal fluid was repeatedly inhibited, so direct whole-genome sequencing was performed, documenting the presence of SARS-CoV-2 sequences.

In July 2022, SARS-CoV-2 RNA was detected in cerebrospinal fluid and urine, and 24 h after these determinations, following approval by the Spanish Drug Agency, a single infusion of sotrovimab (500 mg) was administered with no adverse events. 

Approximately 8 weeks after sotrovimab administration, SARS-CoV-2 RT-PCR on cerebrospinal fluid was negative. In October, stool samples tested positive for SARS-CoV-2. Other samples remained negative.

On 19th December, the patient presented with a new COVID-19 infection and received treatment with nirmatrelvir/ritonavir. He had received a dose of bivalent SARS-CoV-2 vaccine 5 weeks earlier and was receiving 10 mg of prednisone and 100 mg of azathioprine. Follow-up SARS-CoV-2 determinations, including appendix tissue, were all negative apart from a blood sample 8 days after diagnosis.

### 2.2. Genomic Analysis

RNA was purified for whole-genome sequencing (WGS) from freshly inactivated specimens using the Qiagen QIAamp DNA Mini Kit (Qiagen, Courtaboeuf, France). In total, 16 μL of RNA was used as a template for reverse transcription using a LunaScript RT SuperMix Kit (New England BioLabs; Ipswich, MA, USA). Whole-genome amplification of the coronavirus was performed with an Artic nCoV-2019 V4.1 panel of primers (Integrated DNA Technologies; Coralville, IA, USA, https://artic.network/ncov-2019, accessed on 6 July 2023) and Q5 Hot Start DNA polymerase (New England BioLabs). The libraries were prepared using a DNA Prep Kit (Illumina; San Diego, CA, USA), following the manufacturer’s instructions, and quantified with a Quantus Fluorometer (Promega; Madison, WI, USA), before pooling at equimolar concentrations (4 nM) and run on a NextSeq system. Sequencing thresholds for optimal sequence analysis were ≥ 90% genome coverage and >30X depth and number of heterozygous positions < 25.

Positive SARS-CoV-2 specimens (Ct value < 39) were extracted (plasma (11 February 2022), cerebrospinal fluid (6 June 22), and gastric biopsy (26 April 22)). Sequencing reactions from plasma, CSF, and biopsy specimens were replicated four, three, and two times, respectively, and pooled for bioinformatic analysis. Only plasma and biopsy specimens yielded sequences above the quality thresholds (Appendix A), while sequences from CSF did not reach the minimum quality values. During the year, follow-up of the patient’s SARS-CoV-2 RNA by RT-PCR was documented at several points on extra-respiratory samples. SARS-CoV-2 was positive in four determinations of whole-blood samples, two stool samples, and in one sample obtained in urine, gastric biopsy, and cerebrospinal fluid, but attempts to sequence other extra-respiratory specimens apart from the previously described failed (Table 2). The data (fastq files) that support the findings of this study are available at European Nucleotide Archive (ENA) under the project accession number PRJEB64130 [5]. 

The two sequences corresponded to the BA.2 (Omicron) lineage and shared a core of fifty-eight fixed SNPs: forty-five of them corresponded to the BA-2 lineage markers, and the remaining thirteen SNPs were strain-specific. A comparative analysis of the SNPs found in each sequence indicated that they differed in five SNPs (three SNPs were called in the plasma specimen (variant 1) but not in the biopsy, and two SNPs were called in the biopsy but not in the plasma (variant 2)). These findings suggest that we were facing two related variants, which likely emerged from a common parental strain (Figure 1), which had diversified from that parental strain acquiring diversity before the blood specimen was taken (it already showed three additional SNPs) and then along the months between the obtention of the blood and biopsy specimens (sequence obtained from biopsy showed another two additional SNPs). The identification of intra-subject diversity by microevolution can be considered as a proxy to infer virus viability. This inference was also supported by the additional identification of another non-fixed sixteen SNPs, ten of them differentially distributed between variant 1 (eight SNPs) and variant 2 (two SNPs) (Figure 1). An RNA-dependent RNA polymerase (RdRp) gene was fully covered in the sequences (Appendix A).

In December 2022, a nasopharyngeal RT-PCR corresponding to lineage XBB.2 was positive, indicating reinfection with a new strain. 

## 3. Discussion

We describe an immunocompromised patient that presents with indirect data of systemic chronic SARS-CoV-2 infection. Extra-respiratory SARS-CoV-2 determinations were made at the evaluation due to the presence of persistent unspecific symptoms after COVID-19 in 2020, which were compatible with the diagnosis of Long COVID syndrome and the posterior development of bilateral optic neuritis, trying to stablish if a previous SARS-CoV-2 infection could have triggered the development of an inflammatory neuritis. Simultaneous SARS-CoV-2 RT-PCR on respiratory samples and extra-respiratory specimens were made in order to exclude acute COVID-19 in case of positive determinations on blood samples.

SARS-CoV-2 persistence is one of the possible physiopathological mechanisms proposed to be responsible for Long COVID, and different reports have described positive determinations of SARS-CoV-2 antigens or RT-PCR in tissues and blood samples, although virus viability has not been assessed in any of these reports by virus culture, in contrast with immunosuppressed patients [6,7]. 

Different extra-respiratory samples showed SARS-CoV-2 RNA, and whole-genome sequencing corresponded with variant Omicron BA.2. Relevant incidence of this lineage was achieved in our country (Spain) in January 2022, a few weeks before our first evaluation [8,9], suggesting the patient, although immunocompromised, had suffered a subclinical recent infection, and that SARS-CoV-2 had been able to establish a chronic systemic infection in the absence of documented respiratory SARS-CoV-2. Also, the level of antibodies detected on first evaluation were well above the cut-off considered as protective; this fact could reflect a recent infection, and although the patient was immunocompromised, he was able to develop a humoral response but probably without a good neutralization capacity. We did not perform neutralization assays to evaluate functionality, a fact that could have reinforced this observation.

Different reports have demonstrated that SARS-CoV-2 virus can establish persistent infection in immunocompromised patients [10] and that SARS-CoV-2 RNA can be detected in extra-respiratory samples in some patients [11]. These patients present relevant intra-host genetic diversity over time, with several modifications in viral genomes compared to the initial sequence responsible of the infection. This accelerated diversity is one of the main hypotheses to explain how major SARS-CoV-2 variants may have appeared at a population level [12].

For several months, the patient presented positive determinations of SARS-CoV-2 RNA in different extra-respiratory specimens; whole-genomic analysis of two of these samples, taken 2 months apart, showed the acquisition of SNPs, suggesting viral evolution and therefore viability. We were not able to perform viral cultures or subgenomic RNA of these samples, a fact that would have allowed us to reinforce our observation, but the presence of maintained SARS-CoV-2 RNA in different samples and tissues over time and the diversity observed may be considered as a robust subrogate marker of viral viability [13]; although, this fact does not allow us to infer if there is any possible pathogenic effect of this chronic infection. One of the limitations of our study is that we were not able to obtain interpretable genomic sequences of the same samples during follow up. Acquisitions of SNPs in the same compartment, like blood or cerebrospinal fluid, will have been a much stronger observation to translate viral viability. 

The patient received a unique infusion of sotrovimab. There are conflicting data about the blood–brain barrier penetrance of monoclonal antibodies [14], but at this moment, this strategy seemed the best option based on the safety profile of the drug and its long half-life [15]. Extra-respiratory SARS-CoV-2 RT-PCR was negative at approximately 30 and 60 days post-infusion, including cerebrospinal fluid, but positive on stool on day 100.

The patient presented with a new SARS-CoV-2 infection and received a 5-day course of nirmatrelvir/ritonavir, obtaining a fast viral clearance on his upper respiratory tract but presenting later (8 days) positive SARS-CoV-2 determinations in whole-blood samples, probably related to the new strain and not related to previous isolations, but no genomic sequences could be obtained from this sample to corroborate this fact. Over the next months, different samples were negative for SARS-CoV-2 RNA, suggesting this episode may have not established a chronic infection, probably due to an early antiviral treatment, a boosted immunity, and a lower dose of immunosuppressive drugs.

In conclusion, in this immunocompromised patient, SARS-CoV-2 seems to have been able to establish a chronic infection in different extra-respiratory compartments; although, the pathogenic effect and consequences of this persistent infection cannot be elucidated. Prospective evaluation with extra-respiratory SARS-CoV-2 determinations of cohorts of immunosuppressed COVID-19 may offer new insights into the long course of the infection and possible pathogenic effects on different tissues, including the central nervous system. 

## Figures and Tables

**Figure 1 viruses-17-00147-f001:**
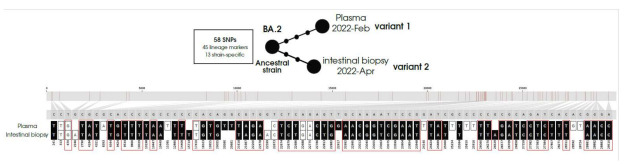
Microevolution of the BA.2 strain in the intestinal biopsy and plasma compartments. Plasma (variant 1) and intestinal biopsy (variant 2) specimens diverged from an ancestral strain. Both specimens share fifty-eight fixed SNPs (forty-five BA.2 markers and thirteen strain-specific) and differ in five fixed SNPs (three SNPs in variant 1 and two SNPs in variant 2). Consolidated (fixed SNPS with a frequency above 80%) are represented by black squares and circles, while unconsolidated SNPs (heterozygous calls) are represented by white squares. BA.2 lineage markers are depicted within red boxes.

**Table 1 viruses-17-00147-t001:** Test results in blood and cerebrospinal fluid sample.

Date	11 February 2022	6 June	11 July	24 August	14 September	18 October	2 December	22 March 2023
**Hemoglobin (g/dL)**	14.5	14.1	14.5	14	13.9	14.4	13.9	14.1
**Leukocyte (10 × 10^3^/µL)**	5.6	4	4.5	7.1	4.7	3.5	3	4.2
**Neutrophil (10 × 10^3^/µL)**	4.8	3.2	3.1	5.7	4.1	2.4	1.7	3.5
**Lymphocyte (10 × 10^3^/µL)**	0.4	0.5	1.1	0.8	0.5	0.7	0.9	0.4
**Platelets (10 × 10^3^/µL)**	176	185	171	182	179	181	167	181
**INR**	1.1	1.15	1.09	1.1	1.13	1.15		
**Fibrinogen (mg/dL)**	353	349	373	348	351	356		
**Glucose (mg/dL)**	103	100	101	80	109	107	76	106
**Serum creatinine (mg/dL)**	0.72	0.83	0.83	0.87	0.87	0.83	0.94	0.75
**ALT (U/L)**	20	19	18	13	14	12	15	14
**Albumin (g/dL)**	4	3.8	4.2	4.1	4.1	4	3.8	4.1
**Ferritin (µg/L)**	71	86	90	59	52	104		62
**Transferrin (mg/dL)**	194	178	196	190	181	176		181
**IST (%)**	45%	38%	52%	39%	45%	46%		34%
**PCR (mg/L)**	<4	<4		<4	<4	<4		
**IL6 (pg/mL)**	3.2	3.1			2.4			<1.5
**IL-1 beta (pg/mL)**	<5	<5			<5			<5
**TNF alpha (pg/mL)**	4.9	6.2			4.4			9.4
**CD3+ T (cells/µL)**	328 (90.7%)	516 (93.6%)	1020 (94.2%)	783 (94.3%)	411 (92.6%)	785 (94.4%	935 (95.6%)	460 (94.4%)
**CD3+/CD4+ T (cells/µL)**	178 (50.6%)	344 (62.2%)	670 (64%)	611 (70%)	275 (62.8%)	576 (68.4%)	655 (67.6%)	305 (62.8%)
**CD3+/CD8+ T (cells/µL)**	136 (38.6%)	176 (31.8%)	322 (30.7%)	210 (24%)	130 (29.6%)	221 (26.3%)	274 (28.2%)	147 (30%)
**CD19+ B (cells/µL)**	19 (5%)	28 (5.1%)	42 (3.7%)	27 (3.4%)	19 (4.2%)	27 (3.3%)	27 (2.7%)	12 (2.4%)
**LGL/CD3-CD16+/CD56+ on NK (cells/µL)**	12 (3.2%)	0 (0.9%)	20 (1.7%)	9 (1.19%)	10 (2.2%)	19 (2.3%)	17 (1.7%)	15 (3%)
**LGL/CD3+CD16+/CD56+ T**	23%		8%	5%	8%	6.7%	6.8%	9.3%
**IgG (mg/dL)**	397	366	446	373	403	406	421	472
**IgM (mg/dL)**	14.9	15.5	16.6	15.7	16	21.9	20.1	24.2
**IgA (mg/dL)**	65.7	47	55	48.3	50	56.8	54	49.2
**C3 (mg/dL)**	86.5	73.3	85.3	77	74	89.6	71.3	81.3
**C4 (mg/dL)**	23.1	20.5	22.9	20.3	20	24.7	19.2	21.8
**Rheumatoid factor**	(-)	(-)		(-)	(-)			(-)
**Serum cryoglobulins**	(-)	(-)		(-)				(-)
**CSF leukocytes (cells/µL)**		0				5		
**CSF red blood cells (cells/µL)**		1330				202		
**CSF proteins (mg/dL)**		49				47		
**CSF glucose (mg/dL)**		59				55		
**ANAs**					(-)			
**Anti-DNA**					(-)			

**Table 2 viruses-17-00147-t002:** Patient evolution: SARS-CoV-2 antibodies and RT-PCR in extra-respiratory and respiratory specimens.

Date	11 February 2022	16 February 2022	4 March 2022	1 April 2022	26 April 2022	6 June 2022	11 July 2022	24 August 2022	14 September 2022	18 October 2022	11 November 2022	2 December 2022	19 December 2022	27 December 2022	19 January 2023	9 February 2023	1 March 20223	22 March 2023
Nasopharyngeal SARS-CoV-2	(-)	(-)	(-)		(-)	(-)	(-)	(-)	(-)	(-)		(-)	(+)	(-)	(-)	(-)		(-)
Whole blood SARS-CoV-2	(+)	(+)	(-)	(-)		(+)	(-)	(-)	(-)	(-)		(-)	(-)	(+)	(-)			(-)
Stool SARS-CoV-2						(+)	(-)	(-)	(-)	(+)		(-)			(-)			(-)
Urine Sars-CoV-2						(-)	(+)	(-)	(-)	(-)		(-)			(-)			(-)
Cerebrospinal Sars-CoV-2						Inhibited	(+)		(-)									
Gastric Biopsy Sars-CoV-2					(+)													
Appendix Sars-CoV-2																	(-)	
Sars-CoV-2 IgG anti-S antibodies	13973 UA/mL					6516 UA/mL	6368 UA/mL	14219 UA/mL	12599 UA/mL	10661 UA/mL		13270 UA/mL	13159 UA/mL		13907 UA/mL			
Sotrovimab administration							(+)											
Nirmatrelvir/ritonavir													(+)					
mRNA Vaccine											(+)							

UA—Arbitrary Units.

## Data Availability

The datasets generated during and/or analyzed during the current study area available from the corresponding author on reasonable request.

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
