# Peer review of "Chronic Systemic SARS-CoV-2 Infection Without Respiratory Involvement in an Immunocompromised Patient"

_viruses, 2025, doi:10.3390/v17020147_

Round 1

Reviewer 1 Report

Comments and Suggestions for Authors

Tejerina F. et al. present a compelling case of a 67-year-old man who had been previously infected with SARS-CoV-2 in March 2020 and subsequently experienced symptoms consistent with long COVID two years after the initial infection. These symptoms were accompanied by the development of bilateral optic neuritis. However, at two years after infection, whole genome analysis revealed that the patient was likely re-infected with the Omicron BA.2 variant, which was prevalent in Spain in 2022. Genomic analyses revealed viral evolution and therefore viability. Approximately 10 months later, the patient contracted another SARS-CoV-2 infection with a different strain, suggesting repeated infections. This well-structured report offers valuable insights into an important aspect of SARS-CoV-2 infection, i.e. viral persistence, whose underlying mechanisms remain largely unexplored. Nonetheless, addressing a few minor points could further enhance the quality and clarity of the manuscript.

1.       It is now becoming clear the impact that persistent symptoms compatible with long-COVID several months, or even years, after initial SARS-CoV-2 infection, may have on various biological processes as well as on everyday life activities. One of the mechanisms underlying long-COVID development is viral persistence (Parumal R. et al, https://doi.org/10.1016/j.isci.2023.106935). The authors should provide a more detailed description of the patient’s symptoms over time, starting from initial infection in March 2020 and considering subsequent infections. For example, did the symptoms improve after the first infection but worsen following the Omicron variant infection? Additionally, given as well the posterior development of bilateral optic neuritis, it would be valuable to investigate persistent peripheral inflammation by means of multiple cytokines detection.

2.       It should be discussed whether the SARS-CoV-2 infections led to severe acute COVID-19 and whether they required hospitalization, all factors which may play a role in the patient’s clinical status.

3.       It would be crucial to address whether other SARS-CoV-2 infections were recorded between March 2020 and February 2022. Multiple re-infections may have a negative impact on the patient’s clinical condition and may provide insights into the progression of symptoms.

4.       The specific moment when the patient came under observation at this particular Clinic should be more clearly described in the text. While some information can be retrieved from the tables, it should be explicitly detailed in the narrative.

5.       The authors should highlight that viral persistence is not limited to immunocompromised patients, and that is a more common situation than initially believed. For example, detection of SARS-CoV-2 RNA in blood has been associated with immune dysfunction in other conditions such as HIV infection (Augello et al. 10.1016/j.isci.2023.108673), but also in non-immunocompromised individuals (Rovito R. et al, 10.1038/s41598-022-23923-1).

6.       The quantification of SARS-CoV-2-specific IgG for the spike protein revealed the presence of antibodies. However, this finding should be better discussed, particularly regarding the antibody levels and their protective capacity. Additionally, it would be important to evaluate whether such antibodies were also functional, e.g. by means of neutralization assays or other available functional assays.

7.       To provide a comprehensive overview of the patient’s clinical status, it would be important to expand a bit more what is known about the lung nodules. May it be a relevant finding in the context of SARS-CoV-2 infection, which is still a respiratory infection?

8.       The histological findings of the gastroscopy performed in April 2022 should be described in more details. What were the clinical concerns or symptoms which led to prescribing a gastroscopy?

Author Response

Cover letter

Dear editor:

We appreciate the assessments made by the reviewers to the manuscript. To fulfill points raised by the reviewers we provide the following explanations and modifications to the original manuscript

REVIEWER 1

1) A more detailed description of March 2020 COVID-19 episode is added to the text to emphasize patient suffered a non-severe clinical curse but maintained later different unspecific symptoms like fatigue, muscle pain, dyspnea, dyspepsia, post-prandial fullness and dizziness. These symptoms fluctuated in intensity but maintained over time:

The patient had been diagnosed of COVID-19 in March 2020 by means of RT-PCR in a nasopharyngeal sample. Clinical curse of the patient was mild, presenting fever and upper respiratory symptoms that resolved in a few weeks without need of hospitalization. After this episode the patient started with fatigue, muscle pain, dyspnea, dyspepsia, post-prandial fullness and dizziness. These symptoms fluctuated in intensity but maintained over time and fulfilled criteria of Long COVID

Also, tests performed at evaluation at another clinical center where optic neuritis was diagnosed are added to the text, to explain that after an extensive battery of studies and a recurrence of optic neuritis the patient was considered to have a Chronic Recurrent Idiopathic Optic Neuritis (CRION), although at that time cytokines determinations were not made:

at another clinical centre, Cervical and cranial nuclear magnetic resonance, immunological assays, and cerebrospinal fluid (CSF) analysis were normal, there were no microbiological findings and anti-aquaporin, and anti-myelin oligodendrocyte glycoprotein antibodies were negative

As stated by the reviewer SARS-CoV-2 persistence is one of the proposed mechanisms associated with Long COVID, but in contrast with immunosuppressed patients where virus viability has been assessed in respiratory samples (but not in extra-respiratory samples), to our best knowledge virus viability has not been demonstrated in Long COVID patients.

To explain this fact a new text has been added in discussion section:

SARS-CoV-2 persistence is one of the possible physiopathological mechanism proposed to be responsible of Long COVID, and different reports have described positive determinations of SARS-CoV-2 antigens or RT-PCR in tissues and blood samples, although virus viability has not been assessed in any of these reports by virus culture, in contrast with immunosuppressed patients [6,7].

With two new references (numbers 6 and 7) the one proposed by the reviewer and another one to highlight that actually viral persistence in Long COVID patients is a hypothesis but there has been no documentation of virus viability

  • Perumal R, Shunmugam M, Naidoo K, et al. Biological mechanisms underpinning the development of Long COVID. iScience. 2023 Jun 16; 26(6): 106935
  • Sigal A, Neher RA, Lessells RJ. The consequences of SARS-CoV-2 within host persistence. Nat Rev Microbiol. 2024 Nov 25

2) Already explained in Point 1 and added to the text, the patient suffered a mild COVID-19 infection and didn´t require hospitalization.

3) The patient did not report any reinfection before evaluation at our center, but as explained in this report we documented that the patient had suffered a sub symptomatic or asymptomatic infection weeks before our evaluation that was able to stablish a persistent infection; but it is impossible to know if the patient had had different reinfections between the first episode in March 2020, and February 2022. The only certitude we have is that SARS-CoV-2 RT-PCR was made at the medical center where the patient was diagnosed of optic neuritis, and it was negative.

To highlight this a text will be added to the manuscript: The patient did not report SARS-CoV-2 re-infections since 2020.

4) To better describe that this patient was first evaluated and followed at another medical center and later in February 2022 started follow up at our center, Hospital General Universitario Gregorio Marañón the following will be added and modified to the manuscript:

A 67-year-old man with a previous diagnosis of chronic recurrent idiopathic optic neuritis receiving immunosuppressant treatment, was first evaluated in our outpatient infectious diseases clinic, Hospital General Universitario Gregorio Marañón, on February 11, 2022

5) SARS-CoV-2 RT-PCR and S or N antigen can be detected in non-immunocompromised patients weeks after the diagnose, but mainly in hospitalized patients with severe COVID-19 and these patients don’t fulfill actually, criteria to what we consider as viral persistence, although these criteria are not clearly validated, and they are focused on immunocompromised patients.

Also, SARS-CoV-2 RNA detection does not stablish viral viability so we consider adding to the manuscript that viral persistence can be developed in immunocompetent patients could confuse readers and persistence of SARS-CoV-2 RNA in critical patients non-immunocompromised should be considered more as part the severe clinical curse than viral persistence as we have defined (with its limitations) actually. So, we consider this hypothesis of viral persistence in patients with Long COVID with previous hospitalization or not is explained with the modifications made in point 1.

6) On first evaluation, February 2022 (unfortunately there is no record of level of antibodies previously), the patient presented levels of AntiS antibodies well above cut-off to be considered as protective levels. He presented 13973 UA/mL (arbitrary units/mL) conversion to BAU/mL 1984,15 (binding arbitrary units/mL). These levels are above the cut-off of 260 BAU/mL that have been considered as protective classically.

These high levels of antibodies could reflect a recent infection and although the patient was severely immucomprompromised he was able to develop a good humoral response, the absence of viral clearance by the patient could translate the lack of functionality of these antibodies but we did not perform any neutralization assay to evaluate this fact.

To clarify this a new paragraph will be added in the discussion section:

Also, level of antibodies well above the cut-off considered as protective could reflect a recent infection and although the patient was immunocompromised, he was able to develop an humoral response but probably without a good neutralization capacity. We did not perform neutralization assays to evaluate functionality a fact that could have reinforced this observation

7) Lung nodules had been documented before March 2020 at another clinical centre and were followed annually with low radiation CT scan in a context lung cancer screening in a patient previously smoker.  A thoracoabdominal CT was made in our centre as part of screening of possible neoplasms that could be associated with the development of optic neuritis. This CT showed 2 lung nodules of 7 mms with no modifications compared with previous tomography, and there were no other findings.

To clarify this, we have added:

There were no significant records in his personal history apart from the presence of two subcentimeter lung nodules, followed at another centre in a context of lung cancer screening. The patient did not report SARS-CoV-2 re-infections since 2020. Thoraco-abdominal computed tomography (CT) performed at our center showed two lung nodules of approximately 7 mm similar to previous CT, with no other findings.

8) Gastroscopy was made due to the gastrointestinal symptoms of the patient: dyspepsia and post-prandial fullness and also to rule out neoplasm associated similar to the indication of thoraco-abdominal CT. There were no macroscopic findings in gastroscopy, so biopsies were made for histological and microbiological evaluation including SARS-CoV-2 PCR. There was no atrophy, metaplasia or dysplasia, Helicobacter pylori was also negative.

To better describe gastroscopy and findings:

In April 2022 due to the gastrointestinal symptoms reported by the patient, a gastroscopy was performed. Tissue biopsies of gastric body and antrum showed no relevant histological findings, without atrophy, metaplasia or dysplasia. SARS-CoV-2 RT-PCR of gastric biopsies was positive

Reviewer 2 Report

Comments and Suggestions for Authors

The association between an immunodeficiency state and the potential risk of persistent SARS-CoV-2 infection, together with the acceleration of viral evolution, has been previously observed.

This study (Case report) by dr. Francisco Tejerina and co-workers “Chronic systemic SARS-CoV-2 infection without respiratory involvement in an immunocompromised patient” describes the presence of SARS-CoV-2 RNA in several extra-respiratory samples in an immunocompromised patient and the absence of virus detection in upper respiratory tract samples. In particular, through whole genome sequencing of a number of samples collected over several months, it was demonstrated the progressive acquisition of different single nucleotide polymorphisms suggesting viral evolution in extra-respiratory sites. 

General comments:

This case report is interesting and further studies in immunosuppressed patients with chronic systemic SARS-CoV-2 infection should be conducted to investigate not only viral viability in different anatomical districts but above all the onset of specific mutations linked to viral compartmentalization.

The title clearly indicates the focus of the article and in the “Introduction” the context of the subject area is adequately addressed. Discussion provides both interpretation of the results in the context of other evidence, and the implications for future studies. Indeed, prospective evaluation with extra-respiratory determinations of SARS-CoV-2 in immunosuppressed COVID-19 patients could offer new insights into the long course of infection and pathogenic effects on different tissues.

The subject is adequate with the overall scope of Viruses, section Coronaviruses.

Minor comments

Introduction:

Last sentence: please specify the acronym SNPs (single nucleotide polymorphisms).

Table 1

Please add the acronym CSF for cerebrospinal fluid in the title.

Table 2 is difficult to interpret and should be better explained in the text. 

Was an intestinal or gastric biopsy performed in April 2022? Even in the text the sample is sometimes described as being of gastric origin and sometimes of intestinal origin.

Genomic analysis and Supplementary Materials

Please specify the acronym WGS (whole genome sequencing).

Suppl Ap. Table 3 as well as Suppl Table 3 are Table S1. Quality values for the sequences obtained.

Please specify the acronym ENA (European Nucleotide Archive).

Author Response

Cover letter

Dear editor:

We appreciate the assessments made by the reviewers to the manuscript. To fulfill points raised by the reviewers we provide the following explanations and modifications to the original manuscript

REVIEWER 2

1) Introduction: Modified to specify single nucleotide polymorphism

2) Table 1: Acronym CSF added

3) Table 2: Text of section genomic analyse describes gastric biopsy as intestinal we modify to gastric to avoid confusion.

To better explain table 2 a new text will be added in the genomic analysis section:

During the year follow up of the patient, SARS-CoV-2 RNA by RT-PCR was documented at several points on extra-respiratory samples. SARS-CoV-2 was positive in 4 determinations of whole blood samples, 2 stool samples and in 1 sample obtained in urine, gastric biopsy and cerebrospinal fluid, but attempts to sequence other extra-respiratory specimens apart from the previously described failed (Table 2).

4) Genomic analysis and supplementary materials: Acronym WGS specified to whole genomic analysis

Suppl Table 3 modified to Table S1

Acronym ENA specified to European Nucleotid Archive

Reviewer 3 Report

Comments and Suggestions for Authors

          The authors describe a case report of a patient with prolonged systemic replication of SARS-CoV-2. Some concerns should be addressed before acceptance of this manuscript.

1.     The Introduction is very limited: it should include more information about the virus, the pandemic, and long COVID, which also may be associated with / due to, viral persistence.

2.     Abstract: it would be more correct to describe the patient on immunosuppressants and not immunosuppressed.

3.     Table 2. Ct values would be preferable after + or -.

4.     Table 2: please define UA.

5.     Page 3, Genomic analysis: please define WGS.

6.     Page 3 and others: presented with. Please add ¨with¨ after ¨presented¨.

7.     Table 2 starts in February 2022, with a negative nasopharyngeal swab. What was the motivation to look for the virus in blood then?

8.     Page 5: the motivation was persistent unspecific symptoms after COVID-19 in 2020? The variant sequenced cannot have been acquired in 2020. Please explain better.

9.     Page 5, end of first paragraph. Instead of documentation, it would be better to use documented.

10. The statement of establishment of a chronic infection is not completely supported by the data presented: continuous positive Ct values are not shown in any of the compartments studied, even in blood, where the virus is more consistently found. The authors should discuss this limitation.

11. In December 2022, a positive nasopharyngeal swab was observed, but the simultaneous blood sample was negative and only positive 8 days later. The authors should discuss this finding.

12. Before the second SARS-CoV-2 infection, Table 2 shows 7 samples positive in different compartments, but only 3 were assayed to produce a complete genome. Why no more blood sample was teste for sequencing? This is a limitation of this case report, and no explanation is given for this. In particular, there is not a follow up Plasma sample to correlate with the SNPs observed in sequence from the biopsy.

13. What was the sequence coverage of the two sequences?

14. Figure 1. What is the meaning of the white spaces? Deletions? If so, not in-frame deletions? What are then the implication for a viable virus?

15. The discussion is also limited, and might include more information on persistent infections and sequence evolution.

16. Date of access to web pages should be included in references.

Author Response

Cover letter

Dear editor:

We appreciate the assessments made by the reviewers to the manuscript. To fulfill points raised by the reviewers we provide the following explanations and modifications to the original manuscript

REVIEWER 3

1) Introduction modified to include more information:

SARS-CoV-2 virus is a betacoronavirus responsible of COVID-19. Few months after the beginning of the pandemic different reports started to describe that some immunocompromised patients presented persistent infections with recurrent pulmonary infiltrates that could be the cause of the death several months after the initial infection [1]. These patients maintained positive determinations of SARS-CoV-2 RT-PCR in respiratory specimens with documentation of virus viability, and in some cases, viral persistence could last for months [2]. In these patients, SARS-CoV-2 can also be detected in extra-respiratory samples and tissues.

Attempts to define persistent SARS-CoV-2 infection in immunocompromised patients have been made, based on microbiological, clinical and radiological criteria. Microbiological criteria are focused on screening for viral presence in respiratory samples and not extra-respiratory samples. Different reports have evaluated the administration of combined antiviral treatment to these patients, achieving in a relevant proportion of these patients viral clearance. [3].

A different syndrome denominated Long COVID develops in non-immunocompromised patients after COVID-19. This syndrome resembles a chronic fatigue syndrome and one the proposed casual hypothesis associated with the development of this clinical syndrome is SARS-CoV-2 viral persistence, although actually this hypothesis has not been validated.

SARS-CoV-2 should be considered a systemic virus with a particular respiratory tropism that could also establish systemic chronic infection in immunocompromised patients in the absence of documented viral infection in nasopharyngeal samples [4].

We report the presence of SARS-CoV-2 RNA in different extra-respiratory specimens in an immunocompromised patient without detection of virus in upper respiratory samples, and without pneumonia or respiratory symptoms. Whole genome sequencing of different samples taken over several months showed the progressive acquisition of several single nucleotide polymorphism (SNPs), suggesting viral evolution in extra-respiratory sites.

A new reference will be added to the introduction.

Reference number 3:

Meijer SE, Paran Y, Belkin A, et al. Persistent COVID-19 in immunocompromised patients-Israeli society of infectious diseases consensus statement on diagnosis and management. Clin Microbiol Infect. 2024 Aug; 30(8):1012-1017

2) Modified to on immunosuppressant treatment

3) Cts added to table 2 in positions where RT-PCR was positive

4) Arbitrary Units (UA)

5) WGS defined as Whole genome sequence

6) With added after the word presented in page 3, 4 and 5

7) The patient suffered persistent symptoms after COVID-19 in March 2020 compatible with Long COVID and approximately one year later was diagnosed of bilateral optic neuritis. Diagnosis was idiopathic optic neuritis, because none of the classical antibodies nor the clinical entities associated with optic neuritis could be detected in a wide etiological study made at another clinical centre. The patient had not received recently any COVID-19 vaccine. Diagnosis of CRION is based in recurrence of optic neuritis episodes and in negative studies with anti-aquaporin and anti-MOGG antibodies but there are other possible clinical syndromes, especially neoplasm, that could be associated with optic neuritis that´s why a tomography was made. Also, we tried to stablish if Long COVID syndrome (this patient started follow up in our centre because of this symptoms) could be related to idiopathic optic neuritis and the patient accepted to be included in a cohort of study of patients with Long COVID where we explore the possibility of SARS-CoV-2 persistence as the cause of this syndrome. In this study we do parallel determinations of SARS-CoV-2 in extra-respiratory and respiratory samples the same day in order to evaluate if the patient is presenting and acute episode of COVID-19 that could be asymptomatic and that we could sequence to stablish if it’s the circulant variant. And in case of a negative respiratory sample and positive extra-respiratory sample we can assume extra-respiratory sample is not related to an acute infection and it could translate persistent infection. We also try to do genomic sequence of positive extra-respiratory to evaluate if they are not circulant variants or if they are variants not represented actually in population but unfortunately due to the low viral load of this samples on many occasions, we are not capable of obtaining interpretable sequences.

This explanation is partially reflected in the discussion section but to expand this information we will add to the text the following:

Simultaneous SARS-CoV-2 RT-PCR on respiratory samples and extra-respiratory specimens were made in order to exclude acute COVID-19 in case of positive determinations on blood samples

8) We think this point is already explained in the discussion section where we explain that prevalence of SARS- CoV-2 variant Omicron BA.2 started in Spain in January 2022, almost 2 years after first documented episode of COVID-19 that was in March 2020.

9) Modified from documentation to documented

10) One of the limitations of this study is that we could not obtained interpretable genomic sequences of all the positive samples obtained during follow up of the patient, mainly because Cts of these samples were all over 30 and the majority of them between 35-40 translating a low viral load of these samples. Our observations will have been stronger if we had been able to obtain longitudinal evolution of the same samples, for example differences in blood samples or in cerebrospinal fluid but unfortunately, we were only able to obtain interpretable genomic sequences of three samples.

To better explain this limitation, we have added Cycle threshold of the different samples in table 2 and the following text will be added to the discussion section:

One of the limitations of our study is that we were not able to obtain interpretable genomic sequences of the same samples during follow up. Acquisitions of SNPs in the same compartment, like blood o cerebrospinal fluid, will have been a much stronger observation to translate viral viability

11) As explained, although several attempts were made to sequence all the positive extra-respiratory samples we didn’t obtain interpretable sequences. We think positive SARS-CoV-2 determination on blood in December 2022 should correspond to the same variant (X.B.B) that caused the new reinfection, but we can´t verify this without genomic sequence, and that probably patient was able to obtain viral clearance later based on the absence of negative repeated SARS-CoV-2 for the next three months.

To better explain this:

The patient presented with a new SARS-CoV-2 infection and received a 5-day course of nirmatrelvir/ritonavir, obtaining a fast viral clearance on upper respiratory tract but presenting later (8 days) positive SARS-CoV-2 determinations in whole blood samples probably related with the new strain and not related with previous isolations, but no genomic sequences could be obtained from this sample to corroborate this fact

12) Explained in the text in section Genomic analysis that we didn´t obtain genomic sequences apart from de 3 samples described in the text: but attempts to sequence other extra-respiratory specimens apart from the previously described failed. Limitations of this point have now been answered in point 11.

13) Sequence coverage is described in Table S1 Suppl Ap. Plasma 93,19. Gastric biopsy 99,05. Nasopharyngeal 98,78

14) The meaning or interpretation of white spaces is there has been no modification compared to the reference allele represented in the grey bar Modifications of the reference allele (SNPs) that are fixed (frequency over 80%) are represented with the black squares and those not fixed (frequency bellow 80%) with white squares, as explained in the legend of the figure.

15) Discussion section has already been enhanced but to broaden information about sequence evolution and possible consequences of this fact in immunocompromised patients with persistent infection the following text will be added to the manuscript:

These patients present relevant Intra-host genetic diversity over time, with several modifications in viral genomes compared to the initial sequence responsible of the infection. This accelerated diversity is one of the main hypotheses to explain how major  SARS-CoV-2 variants may have appeared at a population level [12]

And a new reference will be added (number 12):

Carabelli AM, Peacock TP, Thorne LG, et al. SARS-CoV-2 variant biology: immune scape, transmission and fitness. Nat Rev Microbiol. 2023 Mar; 21(3):162-177

16) Already in the reference section. Reference number 5. 

Round 2

Reviewer 3 Report

Comments and Suggestions for Authors

The authors addressed satisfactorely the concerns.